# Five Year Trends of Particulate Matter Concentrations in Korean Regions (2015–2019): When to Ventilate?

**DOI:** 10.3390/ijerph17165764

**Published:** 2020-08-10

**Authors:** Dohyeong Kim, Hee-Eun Choi, Won-Mo Gal, SungChul Seo

**Affiliations:** 1School of Economic, Political and Policy Sciences, The University of Texas at Dallas, 800 W Campbell Road, Richardson, TX 75080, USA; dohyeong.kim@utdallas.edu; 2Department of Environmental Health and Safety, College of Health Industry, Eulji University, Seongnam 13135, Korea; heen001223@gmail.com (H.-E.C.); 1990134@eulji.ac.kr (W.-M.G.)

**Keywords:** particulate matter, natural ventilation, indoor air quality, regional variation

## Abstract

Indoor air quality becomes more critical as people stay indoors longer, particularly children and the elderly who are vulnerable to air pollution. Natural ventilation has been recognized as the most economical and effective means of improving indoor air quality, but its benefit is questionable when the external air quality is unacceptable. Such risk-risk tradeoffs would require evidence-based guidelines for households and policymakers, but there is a lack of research that examines spatiotemporal long-term air quality trends, leaving us unclear on when to ventilate. This study aims to suggest the appropriate time for ventilation by analyzing the hourly and quarterly concentrations of particulate matter (PM)10 and PM2.5 in seven metropolitan cities and Jeju island in South Korea from January 2015 to September 2019. Both areas’ PM levels decreased until 2018 and rebounded in 2019 but are consistently higher in spring and winter. Overall, the average concentrations of PM10 and PM2.5 peaked in the morning, declined in the afternoon, and rebounded in the evening, but the second peak was more pronounced for PM2.5. This study may suggest ventilation in the afternoon (2–6pm) instead of the morning or late evening, but substantial differences across the regions by season encourage intervention strategies tailored to regional characteristics.

## 1. Introduction

The quality of the air we breathe every day, both outdoors and indoors, is a matter of great concern since it contains various polluting substances which have negative effects on human health and on the environment [1]. According to the 2017 Global Burden of Disease Study, exposure to outdoor air pollution is one of the leading risk factors for premature death, accounting for 3.4 million deaths each year [2]. One of the most critical air pollutants is particulate matters (PM) which are considered as one of major reasons for increased prevalence or exacerbation of respiratory diseases [3,4], cardiovascular diseases [5], and diabetes [6]. The International Agency for Research on Cancer (IARC) has designated the atmospheric PM as a carcinogen of the same class as asbestos and those found in tobacco smoke [7]. However, a growing number of studies have reported that indoor PM could be more harmful in comparison to outdoor air quality because potential air pollutants are built up in confined environments [8,9]. Pollutants are endlessly generated in indoor environments due to human activities and inflow of external air and the purification of contaminated indoor air quality is difficult [10]. Due to a modern lifestyle in which people stay indoors for longer in most industrialized countries [11], the health impacts of indoor PM have been reported to outweigh those of outdoor air pollution, particularly for long-term exposure even at low concentration levels [12,13]. This is particularly relevant for susceptible groups such as children and elderly people who spend most of their time indoors [14,15].

Policymakers and professionals have developed strategies to manage indoor air quality such as improved ventilation and filtration, regulations and standard setting, routine monitoring and inspection, etc. [16]. Natural ventilation has been considered as the most economical and effective means of improving indoor air quality [17], which leads to improved productivity [18], mitigates allergic diseases [19], and even reduces the risk of airborne contagion [20]. Although natural ventilation may be less effective than mechanical ventilation in some settings [21], simply opening the windows for 10 min was found to be effective in achieving satisfactory indoor comfort conditions and air quality [22]. While natural ventilation is widely recommended for indoor air quality, its effectiveness depends on the quality of outdoor air [23,24]. Various concerns have been reported regarding its application, such as ambient particle concentration, indoor source intensity, indoor-outdoor air exchange rate, and circulation of outdoor air within the building, but the largest concern centers on the entry of polluted air from outdoors [25]. Due to the high filtration rate of particles from outdoors, the linkage between indoor and outdoor particle concentrations is more pronounced for a very small particle such as PM2.5 [26]. In particular, residential housing units are highly influenced by the quality of outdoor air since they tend to depend more on natural ventilation even if mechanical ventilation units are installed [27].

Considering a strong connectivity and dependency between indoor and outdoor air quality, natural ventilation could generate negative health impacts where the outdoor air quality is not acceptable for ventilating a building [28]. Despite the importance of risk-risk tradeoff between the exposure to indoor and outdoor air pollutants [29], except for a few governmental reports [17] there is no research that directly provides guidelines for the appropriate time and duration for natural ventilation based on the detailed assessment of temporal trends of ambient air pollutants in multiple communities. As hourly patterns of PM 2.5 and PM 10 could vary due to local characteristics such as major sources of contamination and seasonal weather conditions, an empirical investigation of long-term, site-specific air quality data is critical for developing evidence-based guidelines.

Despite the efforts of the Korean government to improve air quality during the past decades based on several laws and regulations such as the Clean Air Conservation Act of 1991 and the Special Act on the Improvement of Air Quality in Seoul Metropolitan Area of 2003, the problem of air pollution is far from being solved in South Korea. The public’s concerns about the health risks associated with particulate matter exposure have substantially increased recently in South Korea, particularly after experiencing the record levels of outdoor PM 2.5 concentrations in many parts of the country in February and March 2019 [30]. Soon after these events, the national assembly of South Korea passed a series of bills which declared PM air pollution as a “social disaster” and provided emergency measures to tackle the problem, such as mandatory installation of air purifiers in classrooms, distribution of masks to vulnerable groups, and the promotion of low-emission vehicles [31]. It has been reported that Koreans spend more than 20 h a day indoors, particularly the most vulnerable groups such as patients, the elderly, and infants [32]. Although the Korea Ministry of Environment (KME) has provided several guidelines to citizens in response to indoor PM exposure, including ventilation, indoor water cleaning, and indoor air quality monitoring [33], they still lack details concerning how to actually implement the measures.

Only a few research articles have reported the seasonal patterns of particulate matter in South Korea, showing that PM10 and PM2.5 concentrations peak in spring and winter respectively, but both were lowest in summer [34]. However, to our knowledge there is no recent study in South Korea looking at hourly patterns of PM10 and PM2.5 concentrations by season in multiple regions based on long-term data. In South Korea, most people do not ventilate frequently due to Asian dust and air pollution problems attributed to industrial activities in China [35,36]. For this reason, indoor air quality of dwellings in South Korea often becomes worse due to anthropogenic activity including cooking which could lead to increases in the level of PM2.5 as well as NO_2_ [37]. In this regard, KME suggested ventilation for over 30 min three times a day during daytime to improve indoor air quality [33], but this recommendation does not indicate a specific time and lacks tangible evidence of dynamic fluctuation patterns of each pollutant. Therefore, this study aims to analyze the historical records of atmospheric PM10 and PM2.5 concentrations in major cities and regions over the past five years (2015–2019) to assess hourly trends of each pollutant for each season and discuss potential reasons behind the patterns. The findings of this study should provide guidance on the most desirable and undesirable time during the day for not only natural ventilation but also outdoor activities, which could reduce the level of exposure to particulate matter and minimize the adverse effects of air pollution. Moreover, the improved understanding of dynamic trends of PM concentrations and their variations across multiple regions and seasons could help policymakers design a more effective strategy in identifying and monitoring region-specific sources of atmospheric air pollution.

The structure of this paper is as follows: the next section gives an overview of the eight study regions along with data collection and analytic processes. The following sections show the results of the analyses focusing on hourly trends of outdoor PM concentrations by year, season and region, followed by the discussion and conclusion sections that summarize the findings of this study and the implications and limitations of these findings.

## 2. Materials and Methods

The hourly PM10 and PM2.5 concentration data supplied by Air Korea (www.airkorea.or.kr) for KME were obtained for seven metropolitan cities (Seoul, Incheon, Daejeon, Daegu, Ulsan, Gwangju, Busan) and Jeju Island, for the past five years from January 2015 to September 2019. Figure 1 shows the mapped locations of the eight regions in South Korea and Table 1 summarizes the sociodemographic, geographic, meteorological, traffic and other relevant information for each region.

The observations from multiple monitoring stations in each region were integrated to calculate the average; very few observations were missing due to impediment or equipment failures during the period, and these were excluded from the analysis. Statistical analysis software (SAS for Windows version 9.1; SAS Institute Inc., Cary, NC, USA) was used to calculate the geometric means of PM10 and PM2.5 concentrations (µg/m^3^) for each hour of a day (24 time slots) in each region, which were then aggregated by year and season to investigate the annual and seasonal changes and variations. Geometric means were chosen as the best summary statistic for the PM data since their distributions were highly skewed. Four distinct seasons in Korea were classified as: spring (March–May), summer (June–August), fall (September–November), and winter (December–February). A series of hourly time graphs were created to illustrate the trends of PM concentrations over a day for each year, season and region, in order to identify good and bad time slots for natural ventilation. Some additional analysis and discussion followed to explain the potential sources of the patterns, including hourly traffic volume in each region.

## 3. Results

### 3.1. Hourly Trends of Outdoor PM Concentrations by Year

Figure 2 shows the hourly patterns of PM10 and PM2.5 concentrations as an annual average for each year between 2015 and 2019. For both pollutants, there has been a gradual reduction from 2016 until 2018, but they rebounded in 2019. The PM10 concentrations were mostly below the KME’s annual average standard of 50 µg/m^3^ throughout a single day, while the PM2.5 concentrations were way above the recently-strengthened annual average standard of 15 µg/m^3^ during the past five years. Despite some annual fluctuations, the overall hourly patterns for both PM10 and PM2.5 show some similarity throughout the five years; they peaked at 8–11am, declined in the afternoon and rose again in the evening. The second peak in the evening until dawn appears more conspicuous in PM2.5 concentrations, which is due to insufficient air circulation via inversion layer caused by the temperature drop on the earth’s surface during these time ranges [38].

### 3.2. Hourly Trends of Outdoor PM Concentrations by Season

Figure 3 compares the hourly patterns of both pollutants across the four seasons, showing a noticeable seasonal pattern: high in spring and winter and low in summer and fall. Similar to the previous study [34], PM10 concentration was highest in spring while PM2.5 concentration was highest in winter. Regardless of season, both PM10 and PM2.5 peaked between 9am–noon and decayed afterwards. The slope of decline in the afternoon was much steeper for PM2.5 than PM10, except for summer, when both concentration levels appear relatively stagnant throughout a day. The PM10 concentrations during spring and winter were above the KME’s annual average standard of 50 µg/m^3^ particularly between 8am and 2pm, while these were under the standard during summer and fall. PM2.5 concentrations were, however, much higher than the annual average standard of 15 µg/m^3^ during all four seasons.

### 3.3. Hourly Trends of Outdoor PM Concentrations by Season: Regional Variation

Figure 4 and Figure 5 show regional variations of hourly and seasonal trends of PM10 and PM2.5 respectively, including Seoul (capital city of South Korea), six metropolitan cities (Incheon, Daejeon, Daegu, Ulsan, Gwangju, Busan) and Jeju Island. As seen in Figure 4, the seasonal and hourly patterns of PM10 are relatively similar across all regions, exhibiting high concentrations in spring and winter and a peak between 7am and 1pm, but the height of the peak and the declining patterns after the peak seem to vary by region. PM10 concentrations in Busan, Incheon and Ulsan were above the KME’s annual average standard of 50 µg/m^3^ in the daytime period during spring, but its duration was much longer in Incheon (9am to 6pm) than Busan (9am–1pm) and Ulsan (10–11am). During winter, however, only Incheon and Daegu show PM10 concentration above the standard for a relatively short period (10am to noon). As for PM2.5, illustrated in Figure 5, the regional variation appears more noticeable. The PM2.5 concentrations are relatively similar between spring and winter in Seoul, Busan, Incheon, Gwangju, Ulsan and Jeju Island, while the winter concentrations were substantially larger than the spring concentrations in Daejeon and Daegu. For all eight regions, the spring and winter concentrations of PM2.5 were above the annual average standard of 15 µg/m^3^ throughout a given day. However, during the summer and fall seasons, the concentrations were above the standard only during the peak time periods in some regions. Unlike PM10, all regions except for Daegu show two peaks: one in the morning and the other in the evening or night.

The distinct patterns of PM10 and PM2.5 concentrations across the regions is mainly due to different size of particles. Relatively speaking, PM10 is associated with dust on roads, while PM2.5, a very tiny air particulate matter, even smaller than PM10, and relates to emission from vehicles such as aerosol and nitrogen oxide (NOx) [39]. For instance, the hourly trends of PM2.5 concentrations in Ulsan and Jeju Island look quite different from those of other regions, particularly in the afternoon and evening, possibly because of its unique traffic patterns. Ulsan is an industrial city known for factories and plumes leading to male-dominated demographics and workforce characteristics [40]. Thus, considering relatively less traffic congestion during commuting time in the city, air quality might be related more to industrial emissions than local traffic emissions. Jeju Island is a famous tourist destination located at the southern end of South Korea, exhibiting unique demographic, traffic and environmental characteristics. To further investigate the impact of traffic volume on PM2.5 concentration, the traffic volume data were obtained for the same eight regions for three years between 2016 and 2018 and the average amount of traffic was calculates for each hour. Figure 6 shows somewhat similar hourly patterns to the PM2.5 patterns found in Figure 5, beginning to increase at around 6am, fluctuating during the day, and declining at 6pm. This degree of similarity could indicate the level of contribution of traffic emissions on PM2.5 concentration trends, which should vary across the regions depending on site-specific characteristics [31,41]. As mentioned above, the traffic volumes in Ulsan and Jeju Island are substantially lower and flatter than the other regions due to their unique sociodemographic and mobility patterns.

## 4. Discussion

This study examined hourly and seasonal patterns of PM concentrations for multiple regions in South Korea based on the recent five-year data in order to find desirable and undesirable times for natural ventilation due to high levels of outdoor concentrations. Overall, the average concentrations of both PM10 and PM2.5 peaked in the morning, declined in the afternoon, and rebounded in the evening (particularly for PM2.5). The highest concentrations were generally found in spring (March to May) and winter (December to February), but PM2.5 concentrations were more concerning since they were above the KME’s annual average standard of 15 µg/m^3^ during peak times in all four seasons. Thus, it is generally advisable that natural ventilation is recommended during the afternoon (2–6pm) but should be avoided in the morning (9am to noon) or in the late evening (8–11pm). However, substantial differences were observed across the eight regions across the four seasons, suggesting intervention strategies tailored to regional climate and emission patterns. Moreover, the actual decision on whether ventilation is needed at a specific time should be based on a more comprehensive consideration of both indoor and outdoor air quality conditions.

Our results were consistent with previous studies in other countries showing a bimodal pattern with peaks during morning and evening rush hours [42]. This pattern was prominent in many other metropolitan cities such as New York, Los Angeles, Beijing, and London, implying that it was mostly attributed to anthropogenic activity, in particular motor vehicle traffic patterns [43,44,45]. Likewise, the morning peak noticed in this study could be mostly due to enhanced anthropogenic activity during commuting hours, while the afternoon valley is mainly due to a higher atmospheric mixing layer, which is beneficial for air pollution diffusion [46]. The morning peak was relatively more rigorous than the evening risk because, in South Korea, work starting time is relatively standard (around 8 or 8:30 am) but finishing time is more widely distributed due to variability in work schedule. This could partly explain why the PM levels in the evening were somewhat lower than those in the morning in this study.

The seasonal pattern of PM concentrations in South Korea is partly due to its unique geographic and climatic characteristics. The average PM10 concentration was highest in spring mainly because of yellow dust transported from northern China and the deserts of the Mongolian plateau by the prevailing westerlies [47]. The low PM levels during summer time are attributed to rainout and washout processes due to the rainy period as well as frequent typhoons, and rapid air circulation in the fall helps in reducing PM concentrations [24,48]. Generally, PM2.5 can stay longer in the air compared to PM10 due to a smaller size [49]. Local meteorological conditions (very low temperature, low wind speeds, surface layer inversions) and weak wind circulation during winter make it difficult to remove PM2.5 [50]. An increased usage of heating fuels during the winter raises PM levels, and they stay in the air for a longer period of time due to the cold surface of the earth and a low mixture rate [51,52]. This could explain why PM2.5 concentrations, unlike PM10, were higher in winter than spring since finer particles tend to be generated widely by man-made sources of emission such as solid fuel heating in winter [38].

In general, the level of PMs within a big metropolitan city is affected by traffic volume, point source pollution around the city (e.g., power plant), and weather conditions (especially wind direction) by season [53]. Increased fuel consumption for heating during winter could contribute to increasing the level of PM2.5, but other factors may also apply. The high winter concentrations of PM2.5 in Daejeon and Daegu (shown in Figure 4) are found to be associated with region-specific seasonal factors, such as the seasonal rise of bio-incineration via fuel use for household heating and cooking and emissions from petroleum-related industries [54,55] and traffic emission and gas-form pollutants [34]. The studies showed that the high level of PM2.5 during winter season in Daejeon area could be attributed to emission pollutants from two big coal-fired power plants located in the northwest of the city under the influence of the main wind direction during the winter season [56]. Similar patterns of PM2.5 concentrations over winter in Daegu could also be due to its geographic and climatic conditions. During winter, wind blowing from the northwest causes pollutants from the industrial complex located in the northwestern side of Daegu to enter the downtown area, and the particles remain stagnant in the area due to the lack of air circulation in the basin-shaped city [57]. Of course, a number of other geographic, climatic and socioeconomic factors specific to each region may contribute to PM trends as well [23].

## 5. Conclusions

Controlling air pollution is a daunting task in all industrialized countries because so many factors are involved and some of those even cross a country’s border. Eliminating or reducing the sources of emission could bring other types of social conflict or dilemma which would require a long-term effort and investment to be resolved. Natural ventilation could be considered as a simple and low-cost solution to diminish the level of exposure to indoor air pollutants, but only when the outdoor air quality is acceptable. This study emphasizes the importance of thorough monitoring of ambient air quality to be used in providing guidelines for when indoor environments should be ventilated in different region. Despite substantial distinctions across the regions by season, this study provides some general suggestions that the time between 2–6pm is most suitable for natural ventilation but it is not desirable either in the morning or late evening, particularly during spring and winter. It also highlights PM2.5 concentrations surpassing the KME’s annual average standard of 15 µg/m^3^ during peak times in all four seasons and rebounding in size since 2019. Although further studies would be needed to confirm this suggestion and implement solutions in practice, the results of this study can be used as basic information for designing a comprehensive environmental health policy in consideration of dynamic exposure to air pollution.

Of course, it would be ideal to consider simultaneously both indoor and outdoor PM values in order to confirm the suggestions of this study. However, we believe that this study is valid in itself because indoor PM data are not readily available in most of the households and, even if available, the patterns would vary greatly vary according to building structure and lifestyle. This study aims to provide a broad guideline on appropriate times for natural ventilation based on outdoor air quality, particularly where indoor PM levels are unknown, but specific implementation should be carried out by considering indoor-outdoor dynamics pertaining to each building or household. Future study should be directed towards the confirmation of the trends found in this study based on actual experimental data on indoor and outdoor PM concentrations measured in sample buildings in multiple regions, instead of the public aggregated data. In addition, gathering more data on covariates indicating geographic, sociodemographic, climatic, industrial and traffic patterns in each region would enable multivariate analysis and modeling for the determinants of spatiotemporal and seasonal changes in PM concentrations. In spite of limitations and the inability to explain some patterns, this study can inform the public regarding the desirable or undesirable times for natural ventilation as well as outdoor activities in each season and region. Moreover, it can encourage policymakers to design season-specific environmental management strategies and practices tailored to regional climate and emission factors such as regional traffic monitoring networks and air quality alert systems for ventilation, which may promote a cost saving, by specialization of intervention, and enhance policy outcomes.

## Figures and Tables

**Figure 1 ijerph-17-05764-f001:**
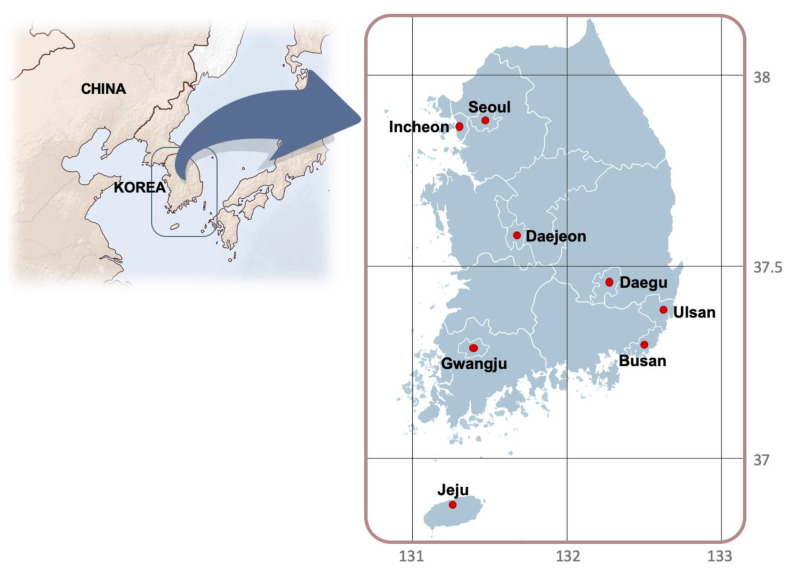
Location of the study regions in South Korea.

**Figure 2 ijerph-17-05764-f002:**
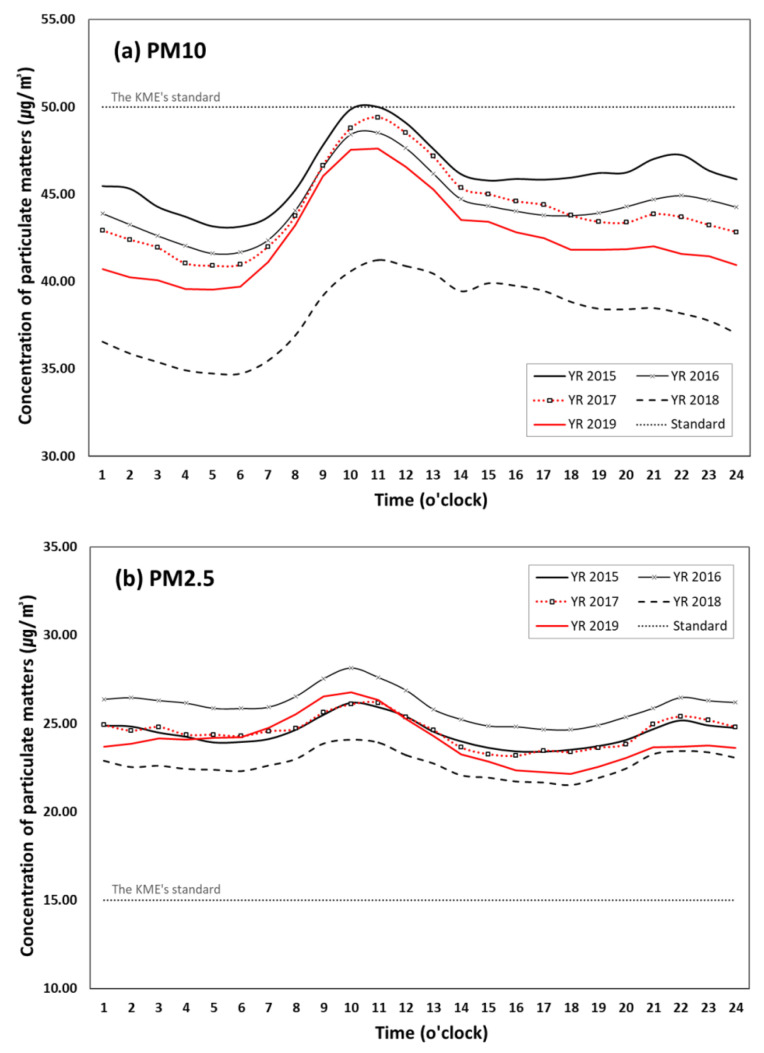
Diurnal variation of particulate matter (PM) concentrations (µg/m^3^) by year (all regions).

**Figure 3 ijerph-17-05764-f003:**
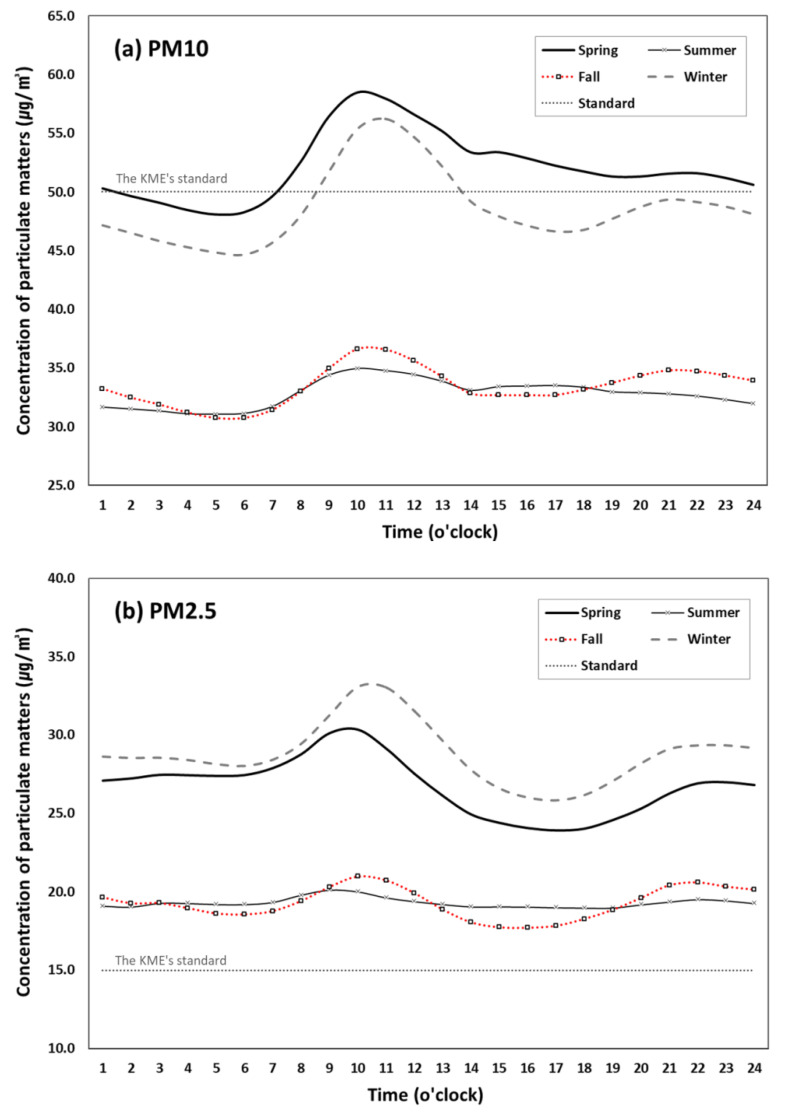
Diurnal variation of PM concentrations (µg/m^3^) by season (2015–2019).

**Figure 4 ijerph-17-05764-f004:**
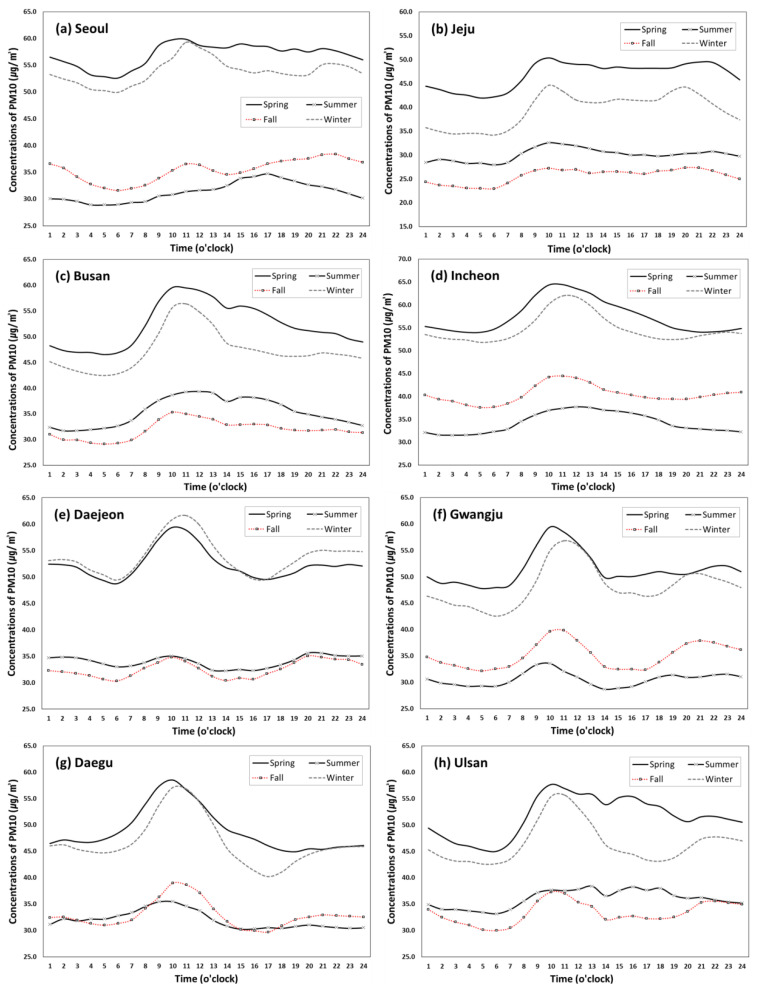
Diurnal variation of PM10 concentrations (µg/m^3^) by season in each region (2015–2019).

**Figure 5 ijerph-17-05764-f005:**
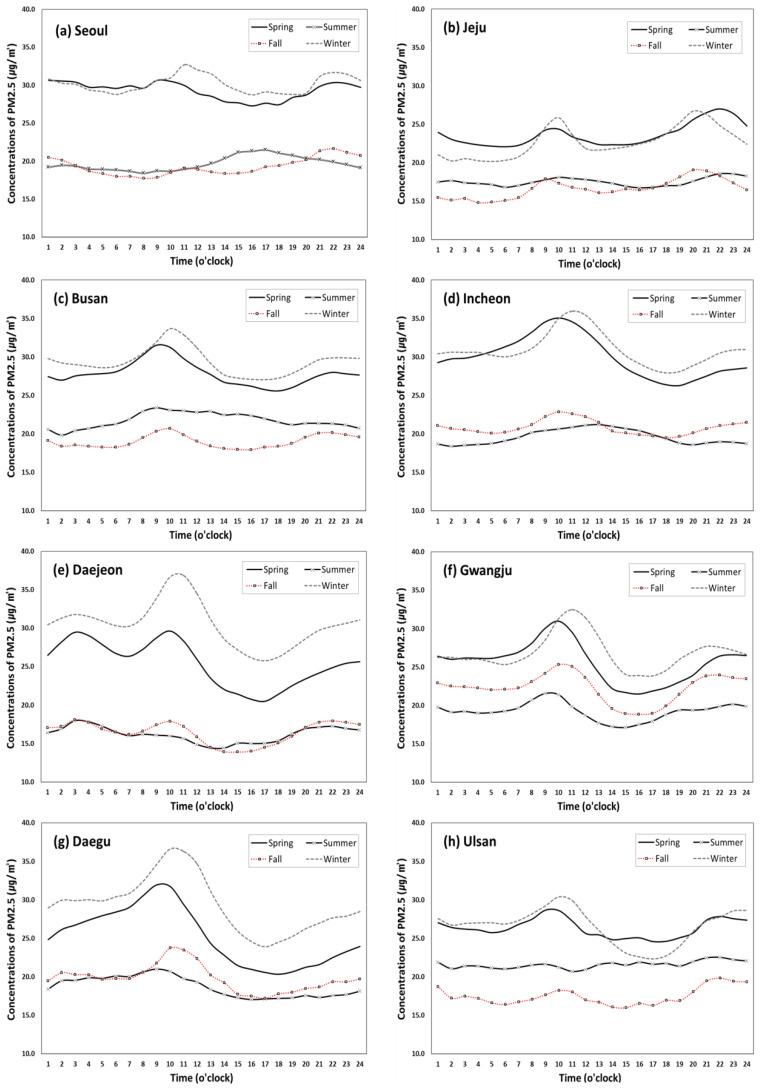
Diurnal variation of PM2.5 concentrations (µg/m^3^) by season in each region (2015–2019).

**Figure 6 ijerph-17-05764-f006:**
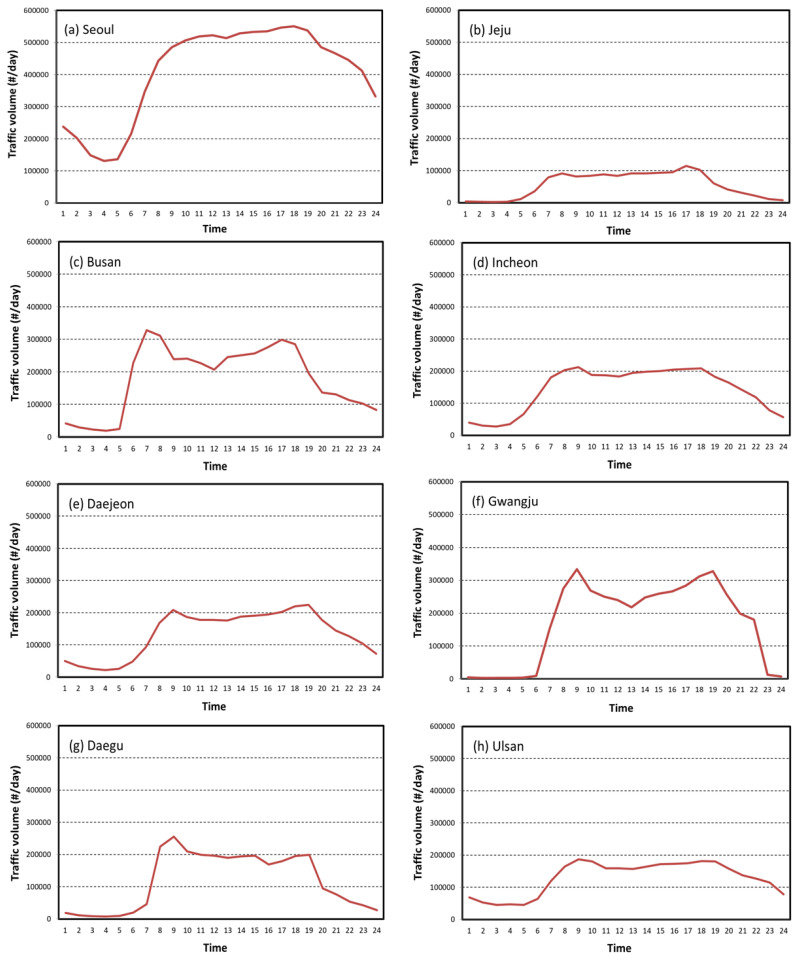
Diurnal variation of traffic volume in each region during 2016 and 2018.

**Table 1 ijerph-17-05764-t001:** Geographic, climatic, socio-economic and other notable characteristics in the eight regions.

Region	Area (km^2^)	Population Density (people/km^2^)	Average Temperature (°C)	Annual Average Precipitation (mm)	Number of Monitoring Stations	Notable Characteristics (Geographic, Meteorological and Mobility Patterns)
Seoul	605	15,964	13.5 (−10.9–36.8)	891.3	25	Capital of South Korea; located in the northwest of the country; heavy traffic congestion during rush hours
Busan	770	4380	15.7 (−4.4–35)	1623.2	25	Located in the south; relatively warm; close to the sea; heavy traffic congestion during rush hours
Daegu	883	2753	14.8 (−7.2–36.9)	995.7	15	Located inlands in south-central region; a basin-type city; high summer temperatures and frequent heat waves; heavy traffic congestion during rush hours
Incheon	1063	2769	13.2 (−10.4–36)	919.5	20	Located in the west; close to the sea; heavily influenced by the northwest wind; contains an industrial complex; heavy traffic congestion during rush hours
Gwangju	501	2980	14.7 (−5.9–34.8)	1085.9	9	Located in the southwest; heavily influenced by the northwest wind in winter; heavy traffic congestion during rush hours
Daejeon	540	2796	14 (−9.6–36)	984.2	10	Located around the center; a basin-type city; power plants around it; heavy traffic congestion during rush hours
Ulsan	1062	1080	14.9 (−5.3–35)	1045.1	17	Large-scale industrial complexes; close to the sea; distinct traffic patterns during rush hours
Jeju	1850	356	16.8 (1–35.4)	1979.9	6	An island city located in the southernmost part of Korea; a large number of tourists and no industrial facilities; constant traffic due to tourism

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
