# Peer review of "Five Year Trends of Particulate Matter Concentrations in Korean Regions (2015–2019): When to Ventilate?"

_ijerph, 2020, doi:10.3390/ijerph17165764_

Round 1

Reviewer 1 Report

The paper presents and highlights the need for conducting research related to an essential aspect of risk-risk tradeoffs for indoor and outdoor dynamics in air quality we are exposed. The manuscript presents the preliminary data analysis, and the potential need for conducting research is the indoor air quality natural ventilation precautions for healthy breathing. However, the paper requires some significant considerations, which I have specified below:

  1. The article discusses the need for indoor-outdoor dynamics, yet the cases are presented based only on the outdoor station values. It would be of great benefit if the authors detailed the rationale of why they have considered only outdoor station values while addressing the indoor air quality as the central theme in more detail? 
  2. The paper compares various locations and the mean value pattern. Still, considering the flow, authors should consider providing the information about the study area like geographical context and meteorological factors, before comparing it. Authors have highlighted that information briefly in the discussion section, but for readers to understand the graphs and their relevance, it is more useful to give context before. 
  3. Authors used geometric means to make their case, more detail about why only that and not some other parameter was considered? A more detailed explanation of why it makes sense to only use that will be useful for readers.
  4. How the structure of the paper looks like in the upcoming section is not conveyed in the article. I suggest the authors consider doing the same to present the flow of text inconvenient way.
  5. Figure 3 and 4 is not at all readable. Furthermore, it will be of great use if the authors also add the standard values line in the graph so that the comparison between the standard value and the fluctuating values are clear. (Line 137-138)
  6. There are a few grammatical and spelling mistakes in the sentences. Example: line 139 
  7. The discussion section presented some of the details of the pattern identified in the previous section; however, it would be relevant if authors considered reshuffling some text to make more sense of the information presented in both the sections. 
  8. It would also be great if the authors can show the traffic information with pollutant values to understand the similarity in trend. 
  9. The authors also missed presenting the limitations of the study with clarity. I recommend giving those explicitly.
  10. A detailed discussion about the spatiality, the morphology of the study areas, and responsible parameters for change in values of pollutants should be described with more details as right not it seems to be in the fuzzy order and details.

Overall author needs to reconsider some significant changes to make the contribution more concrete and useful for the domain.  

Author Response

Please see the attachment. Many thanks for your comments.

Reviewer 2 Report

This study examined hourly ambient PM10 and PM2.5 patterns in seven metropolitan areas and one island in South Korea, with the goal of determining the most beneficial times for building ventilation. The authors found the lowest concentrations in the afternoon and higher concentrations in the late morning, which partially contradicts government guidance and may have policy implications. This paper would be strengthened with more discussion of indoor PM2.5 sources and concentrations in this setting and comparisons of the observed ambient PM hourly patterns with other cities in the region.

Major comments:

  1. Introduction, lines 35-37: Please update the GBD estimates with the 2017 GBD report: https://doi.org/10.1016/S0140-6736(18)32225-6
  2. Introduction, lines 41-43: It would be helpful to provide a citation for the statement that people are staying indoors longer. Does this apply globally or regionally?
  3. Materials and Methods – please provide more information on how data from multiple stations were integrated (mean? median?) and how you handled missing data. If one day was missing was data for the entire 5 year period dropped?
  4. The analysis of traffic patterns in the Discussion section is superficial and only moderately correlated with the PM trends. Please move the analysis of traffic patterns to Methods and Results as a main aim of the paper, or remove it.
  5. There are other papers which have analyzed hourly trends in ambient air pollution in other metropolitan areas and found similar patterns. While you state there are no recent papers which look at hourly concentrations in South Korea, it would be helpful to compare the observed hourly patterns with literature from other cities in the region or elsewhere.
  6. A major problem with the paper is that the authors conclude that ventilation in the afternoon is beneficial, however we have been given no information on indoor PM concentrations in the study region. While there are major sources of indoor PM in many parts of the world (tobacco smoking, biomass cookstoves), it would be useful to know about major sources of indoor PM2.5 in the study area. Perhaps indoor PM concentrations are low enough that ventilation is never advised, since ambient PM2.5 concentrations are consistently above guidelines. Another possibility is that there are indoor gaseous pollutants (NO2 from gas stoves, VOCs, etc) which should be ventilated. However, these pollutants were not examined in this study and it is difficult to assume ambient concentrations of gaseous pollutants correlate with ambient PM2.5 during the day. I would recommend the authors 1) add more discussion of existing literature on concentrations and sources of indoor PM2.5 in the study region 2) change the language throughout the paper to emphasize the results (ambient PM2.5 is lowest in the afternoon compared to other times), without making firm statements that ventilating indoor spaces during the afternoon is beneficial.
  7. While this study focus on ventilation, it would strengthen the paper to discuss the implications of hourly ambient PM2.5 trends on other outdoor activities, such as exercise or school activities.
  8. One very interesting and policy-relevant finding of this study is that the KME recommendations for ventilation (10am-9pm, Introduction line 91) include the hours with highest concentrations in this study. It would strengthen this paper to explicitly compare this paper’s results with the KME recommendations in the Discussion section.
  9. Please improve the resolution of Figures 3, 4, and 5.

Minor comments:

  1. Figures: Please increase the font size of all text and labels in the figures.
  2. In the reference list, please provide more information for references 29, 33, and 43.

Author Response

(The authors gave the same response as above.)

Reviewer 3 Report

Title:

 5-year trends of particulate matter concentrations in Korean regions (2015-2019): when to ventilate?

Auhors:

 Dohyeong Kim 1,*, Hee-Eun Choi 2,*, Won-Mo Gal 2 and SungChul Seo 2,

[1] General explanation on a review result by reviewer:

The authors tried to suggest proper time when the natural ventilation of indoor air for economically and effectively means of improving indoor air quality by outdoor air should be taken in 8 cities in the South Korea.  The analysis of hourly patterns of PM10 and PM2.5 was also well in detail, explained why their concentrations were different in each city for each season with different regional characteristics like industry and touristic places.   

-- Thus, this paper is qualified to be published in this journal, if all figures should be re-drawn clearly and with minor correction and proper explanation is given to questions suggested by reviewer.   

[2] Question and Corrections:

-- The authors well explained typical characteristics of

(1) In hourly patterns of PM10 and PM2.5 as an annual average for each year from 2015 to 2019 for all regions, the author explained that peak concentrations were in 8~ 11AM, declined in the afternoon, and second peak in the evening due to insufficient air circulation under inversion layer.

-- Question 1)  If possible, please explain why PM10 and PM2.5 concentrations were peak in the morning and declined in the afternoon.

(2) In seasonal averages for 5 years (2015~2019) for all regions, PM10 concentration was highest in spring due to dust , but PM2.5 was in winter.   I agree that peak concentrations of hourly PM10 and PM2.5 were in 9 ~ noon and afterwards decayed over all.

(3) In hourly patterns of PM10 in spring, summer, fall and winter each year in 8 cities, I agree that generally higher concentration took place in spring due to yellow dust transported from northern China with relatively larger size of particulate matter.   In winter, most of cities is under in the rare influence of dust transported from China and is in the emission of relatively smaller size of particulate matter and gases from both the increased usage of heating fuel and traffic vehicles on the road.                

-- Question 2) For instance, the authors explained that higher PM2.5 concentration over winter in Daegu city was due to its geographic and climatic condition. For easily understanding by reader, please explain more clearly what the meanings of geographic and climatic condition are with a short sentence, if possible.  

(4) Fig. 5 of traffic volume was very useful for easily understanding Fig. 4.

-- Question 3) The authors said that the traffic volumes in Ulsan and Jeju are lower and flatter than the other regions due to their unigue sociodemographic and mobility patterns.

Please explain What their unigue sociodemographic and mobility patterns were.

(5) There are some mistyping of words such as unieque (225 line).  The authors should read again carefully and modify those words.   

(6) Requirement of Figure correction:

Please draw all figures such as Fig. 1, Fig.2, Fig.3, Fig. 4 and Fig. 5.,  

For instance, as both characters in x- and y- axis (Concentration, ---, 30, 35,   Time, 1, 2, ) and years (2015, --,) inside a small box is not clear, please make them thick and use slightly larger characters.

Author Response

(The authors gave the same response as above.)

Round 2

Reviewer 2 Report

The authors have sufficiently addressed all comments.